# Protection of Environmental and Natural Values of Urban Areas against Investment Pressure: A Case Study of Romania and Poland

Paulina Legutko-Kobus [1] , Maciej Nowak [2] , Alexandru-Ionut Petrisor [3,4,5,6,*] , Dan Bărbulescu [7],
Cerasella Craciun [3,8] and Atena-Ioana Gârjoabă [3]

1   Department Public Policy, SGH Warsaw School of Economics, 02-554 Warsaw, Poland
2   Department of Real Estate, Faculty of Economics, West Pomeranian University of Technology,
    70-310 Szczecin, Poland
3   Doctoral School of Urban Planning, Ion Mincu University of Architecture and Urbanism,
    10014 Bucharest, Romania
4   Department of Architecture, Faculty of Architecture and Urban Planning, Technical University of Moldova,
    2004 Chisinau, Moldova
5   National Institute for Research and Development in Tourism, 50741 Bucharest, Romania
6   National Institute for Research and Development in Constructions, Urbanism and Sustainable Spatial
    Development URBAN-INCERC, 21652 Bucharest, Romania
7   Doctoral School in Ecology, Department of Systems Ecology and Sustainability, University of Bucharest,
    050095 Bucharest, Romania
8   Department of Urban Planning and Landscape, Faculty of Urban Planning, Ion Mincu University of
    Architecture and Urbanism, 10014 Bucharest, Romania
*   Correspondence: alexandru_petrisor@yahoo.com; Tel.: +40-213-077-191

**Abstract:** Although conservation and development are two facets of sustainability, they are often placed in contradictory positions. In this context, planning systems are able to respond to investment pressure, especially in countries with underdeveloped institutional solutions for this purpose, and are consequently characterized by a shifting relationship between spatial planning and environmental protection. Although these issues have been relatively well conceptualized, the literature still lacks more in-depth analyses of selected case studies. In order to fill the gap, this study aimed to identify potential ways to protect the environment and natural values in urban areas from investment pressures in countries with less developed planning systems, based on a comparative Polish-Romanian perspective. The method consisted of comparing the national legal frameworks for environmental protection and spatial development and analyzing in detail two case studies from each country. The findings indicate that national protection is required in both countries to ensure the effective protection of natural areas situated within city administrative limits that provide important ecosystem services. Moreover, the results reveal the need for more research on similar areas using multi-scale interdisciplinary approaches and reviewing planning theory with respect to its efficiency in protecting nature.

**Keywords:** opportunity cost; urban natural protected areas; biodiversity conservation; urban sustainability; transnational comparison; Băneasa Forest; Vacaresti Natural Park; Kabacki Forest; Bielanski Forest

## 1. Introduction

### 1.1. Background

Sustainable development presumes safeguarding today's biodiversity in natural protected areas for future generations, while continuing socioeconomic development in a responsible manner for the environment and society [1]. However, this is hard to achieve in practice, and conservation and development often oppose each other [2]. The opposition is even stronger in natural protected areas where land resources are scarce and opportunity costs often underestimate the value of natural heritage [3].

Developing a response to investment pressure (e.g., pressure from investors on public authorities aimed at implementing the development they want) is a key challenge of any national spatial planning system, especially in urban and suburban areas where the pressure is higher. Planning systems should protect diverse site values, i.e., environmental, natural, cultural, compositional, and aesthetic values, and also the ability to support urban life in an integrated and healthy manner. The necessary scope of protection differs across countries. Investment pressures also differ among countries, especially in relation to particular types of land. However, there is no doubt that the lack of effective institutional solutions in this respect results in spatial chaos, loss of some natural, anthropic, and particularly local cultural heritage values, and additional social costs. For this reason, some authors consider that this form of urban planning can have an ambiguous character [4]. The problem is particularly significant in planning systems that have underdeveloped institutional solutions in terms of protection against excessive investment pressure. Romania and Poland undoubtedly belong to this group, sharing a common history including the experience of communism, with a significant impact on spatial planning. Property ownership is a very strong manifestation of post-communist "rebirth" in spatial planning systems. Therefore, it seems particularly important to look for solutions that, at least to some extent, provide protection and preserve unique specific heritage values through planning, in tight relation with landscape quality and in particular, integrated urban morphology. Analyzing specific case studies, reasons, and consequences of certain solutions and practices can provide discussion to help improve poorly functioning national spatial policy systems. Interdisciplinary analyses of these issues from different perspectives are crucial.

Boulton et al. [5] approached the issue of urban green space in a broader way, also referring to the role of urban planning in the context of urban nature conservation. These authors identified several research directions requiring further analysis, i.e., (1) the impact of system certainty/flexibility on achieving specific objectives, (2) the search for new solutions based on case studies, and (3) a broader reference to multi-level governance. The first two topics relate to the discussion of flexibility in urban planning and endangering the health of the urban organism, which directly influences human metabolism [6–9].

A second important point of reference is the comparison of conservation area solutions in Central and Eastern European countries. Yakusheva [10] considers this an important issue, pointing out many common features of the countries indicated. In this context, she points to similar problems stemming from the communist era, barriers regarding Europeanization, and similar rigid legal frameworks also confirmed by other studies [11]. From the perspective of the topic addressed in this article, it is also necessary to add the weaknesses of national spatial planning systems in both countries, doubled by rigid or far too permissive planning processes. These weaknesses significantly hamper the overall comparison of both systems with those in Western Europe [12].

### 1.2. Specific Issues

Spatial problems caused by urban growth are often noted and analyzed in the literature. Extensive, often uncontrolled, urban growth is found in many parts of the world [13–18]. Solly [19] points out the challenges faced by contemporary planners in devising limits to urban sprawl (similarly [20]). A manifestation of the above is the major negative transformation of suburban landscapes [21]. We can also add here the uncontrolled, excessive development of particular parts of cities. The consequences are diverse. On the one hand, there is serious spatial chaos, generating very high costs [22]. On the other hand, limited accessibility to public spaces [23] and social constraints and barriers appear to be increasing to an extent [19,24]. Often, the actions of specific municipal authorities, focused on maximum profit, prevent effective attempts to curb the indicated trends [25]. It is worth referring to the concept of Right to the City, which, especially nowadays, is particularly relevant [26–28], in addition to the Right to Landscape and Landscape Right [29–31].

The relationship between spatial planning and environmental sustainability has been described as a shifting one [32]. Currently, urbanization occurs at a very fast pace in the

form of two main trends: the compact city (through densification of urban fabric) and urban sprawl, both representing potential environmental dangers [33]. The main problems affecting the environment, i.e., land cover and use changes, climate change, and alterations of energy flow, have been merged under the concept of "global changes" [34], with deep planning implications [35]. Wise planning can regulate land cover and use changes, mitigating the effects of climate change [36,37] and providing normal functioning ecosystems, able to ensure the self-regulation of energy flows and, implicitly, supporting the conservation of biodiversity [38]. The fragmentation of ecosystems and anthropogenic land use changes are major threats to biodiversity conservation [39], determined especially by urban sprawl, and constitute a common European feature [40]. Ultimately, wise planning contributes to global sustainability. The level of integration of planning and environmental protection varies across the world, from (1) total separation to (2) integration via legislation, and ultimately (3) through institutional structures. Petrişor and Petrişor [32] classified countries based on their 2013 governmental structure; the first category included Germany, the United States, Canada, China, Japan, while the second category included Romania and Poland. Examples of institutional integration are found in France (Ministry of Sustainable Development including departments dealing with environment, energy, transportation, and spatial planning), Argentina (Ministry of Planning, Public Works and Services including departments responsible for energy, planning, public works, transport, and communication), and Italy (Ministry of the Environment, Territory, and Sea). However, institutional structure does not necessarily indicate the importance of environmental issues. Nevertheless, in Japan, planning and environmental issues are addressed by different ministries with environmental constraints prevailing over the others [41]. Such right of way rules are important when dealing with land, as different uses correspond to different opportunity costs [42,43] and are able to generate conflicts between stakeholders with different interests [44], e.g., the private sector, especially when conservation is an option [3,45].

Very often, the liberalization of spatial planning results from a broader consideration of individual investor perspectives by public authorities. Sometimes the specifically understood notion of planning flexibility serves this purpose [46]. While in principle this is not wrong, it is important to remember that investors are one of many groups interested in shaping space, and therefore, other interests may be considered too. The literature provides examples of unambiguously negative practices [47]. With this caveat, we agree that spatial planners should understand well how large investors operate [48]. Raco et al. [49] pointed out that the perspective of developers is often presented in a caricatural, oversimplified way. For this reason, the relationship between the real estate investment process and urban planning [50], and the rationale by which investors are more considerate to environmental requirements [51], requires extensive analyses. Taking such perspectives into account and integrating them in a broader concept of spatial policy (including reconciliation with other objectives) can have positive effects [52] and increase planning efficiency.

There are numerous proposals for adapting and changing spatial policies to counteract the indicated trends more effectively. The starting point would be to seek broader protection of diverse values, e.g., environmental and social [53]. The following lines of discussion can be identified here:

- Correctly defining spatial challenges, boiled down to reconciling different interests and perspectives [54];
- Understanding (at the spatial policy level) the reasons for wider urban pressures in a particular area [55,56];
- Adapting spatial planning to new approaches that guarantee better efficiency when protecting valuable space assets [57–61];
- Rethinking new contemporary theories and/or the application of integrated and resilient complex urban planning theories applied to real case studies [62];
- Comparative studies of problems and responses of national spatial planning systems [63–67];

- Role of public authorities, including permissible scope of interference in the market [68–70];
- Increasing the resilience of cities [70]. Heurkens et al. [71] argued that this goal is achieved through an effective combination of planning policy instruments and market-driven actions.

The indicated issues relate to a separate issue, i.e., linking environmental and nature conservation with spatial planning, regardless of the effective operation of other environmental tools [72]. The goal of spatial planning should be sustainable natural systems able to supply real human needs [73], taking into account socio-economic systems specific to the local context [74]. This can translate into shaping specific development zones [75]. Climate protection priorities should also be highlighted in this context [76]. These objectives should be applied in both strategic [77] and regulatory planning instruments.

The challenges identified also apply to urban areas. Papargeorgiou and Gemenetzi [78] pointed to the particular role of green spaces and parks, which can be categorized in different ways, resulting from their multiple ecosystem services [79]. This perspective also appears in other studies [80]. The relationship between urban areas and the environment is very important. Urban expansion and associated land use changes contribute to environmental pressures in other sectors [81]. Paying more attention to environmental and natural issues in urban development means redefining urban policies in many cases [82].

Discussion of the indicated topics points to the optimal scope of spatial development plans in response to environmental challenges. Zwirowicz-Rutkowska and Michalik [83] emphasized the need to include many environmental protection principles in plans. Gonzalez [84] pointed out the variation in scope and roles of land use plans in different countries. Above all, it is important that environmental planning provisions are underpinned by environmental analyses, based on the cooperation of various participants in the land use planning process [85,86]. Spatial policy instruments should never fail to protect the environment and naturally valuable areas [87,88] or generate spatial conflicts themselves [89,90]. The reasons for the above are the separate, sectoral treatment of environmental issues and spatial planning [91].

Today, it is necessary to reanalyze the possibility of developing growing but truly green cities, taking into account planning theory and the contradictions of sustainable development [92]. A fundamental reevaluation of the theory of interactions between natural environment issues and urban planning processes is also needed to develop sustainable and resilient urban management by re-appraising and re-assessing with the new tools used in urban and territorial planning [93].

The literature includes detailed accounts of national spatial planning systems that attempt to relate their problems to the international discussion [3,4,12,17,41,71]. There are also general characterizations of spatial planning systems in many countries [10,19,38], e.g., those of Central and Eastern Europe [64]. However, due to the large number of countries compared, it is not possible to include all detailed aspects, including those related to nature conservation. Comparisons between two specific countries are also made. An example of this is the comparison of green infrastructure issues in Romania and Poland [79]. This article takes up further threads comparing the nature-spatial issues in the two countries.

The above literature review covers relevant issues from the perspective of the two compared countries and part of a broader international discussion. The detailed systemic issues of the two countries, including differences, are covered later in the article. At this stage it is worth pointing out that the indicated issues are also noticeable in large cities of Romania and Poland. These issues include urban sprawl, significant investor pressure threatening environmental and natural assets, and limited access to public spaces.

In order to fill the identified knowledge gap, this study aims to find potential ways to protect the environmental and natural values of urban areas from investment pressures in countries with less developed planning systems. The text includes an interdisciplinary urban planning analysis in the cities studied. We consider that a detailed analysis of the planning conditions in the indicated areas of cities would allow for a broader identification

and comparison of the solutions presented. The article makes an important contribution concerning conservation comparisons in large cities showing how, despite different legal and institutional frameworks, nature can be protected by focusing on naturally valuable areas under investment pressures. The article is comprehensive, deliberately detailing and comparing selected case studies. In addition to the scientific dimension, it also has a significant practical dimension aimed directly at municipal authorities.

## 2. Methods

This article explicitly compares Romania and Poland in terms of their national legal frameworks for environmental protection and spatial development, based on choosing and comparing two representative case studies from each country. As indicated by previous studies looking at planning for green infrastructure [36,79] or the general planning systems [64] of the two countries, Romania and Poland have a shared communist period with a similar dynamic that differs from that of other Eastern-European countries, followed by a recent accession to the European Union, and their political ties make their comparison not only possible, but also relevant for other similar countries, especially those with emerging economies [36].

The methodology consisted of several phases (Figure 1). In the first phase, the paper reviewed international literature about investment pressures on spatial planning and ways and scope of environmental and nature protection. Then, the second phase characterized Romanian and Polish solutions. We analyzed in detail case studies able to show how protection against investment pressure was ensured, from the institutional perspective and theses of literature on the subject. On the basis of the indicated analyses, conclusions on the direction of change were proposed.

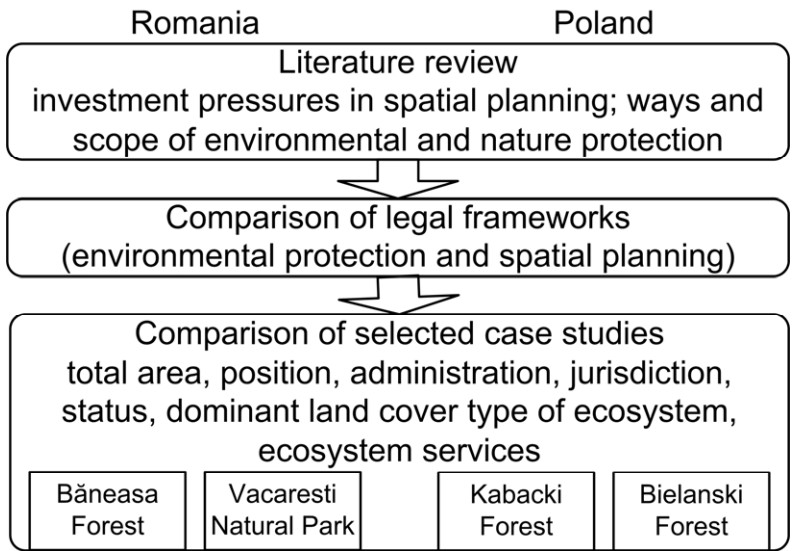

**Figure 1.** Phases of the methodology used in the present study.

In order to present a practical approach, 4 case studies were selected in the second phase, including two in Romania and two in Poland. All case studies were located in the capital cities of Bucharest and Warsaw. The location of the countries and cities in Europe is presented in Figure 2. Since the case studies were small areas, the image shows only the location of cities where the case studies were located. The four analyzed sites were characterized, indicating which natural functions and ecosystem services they provided in the city. In a second step, examples from each country, i.e., Băneasa Forest and Vacaresti Nature Park in Romania and Kabacki and Bielanski Forests in Poland, were characterized in more detail, indicating what recommendations and constraints were formulated in the planning documents. The characterization of their land cover and use relied on the most

recent Urban Atlas data from 2018, available free of charge from the Copernicus Program of the European Union (https://land.copernicus.eu/local/urban-atlas/urban-atlas-2018, accessed on 30 October 2022). Maps were produced using ArcView GIS. Whenever necessary (e.g., analysis of Băneasa Forest), additional data were obtained from the National Office of Cadastre and Real Estate (https://geoportal.ancpi.ro/portal/home/webmap/viewer.html?useExisting=1, accessed on 30 October 2022).

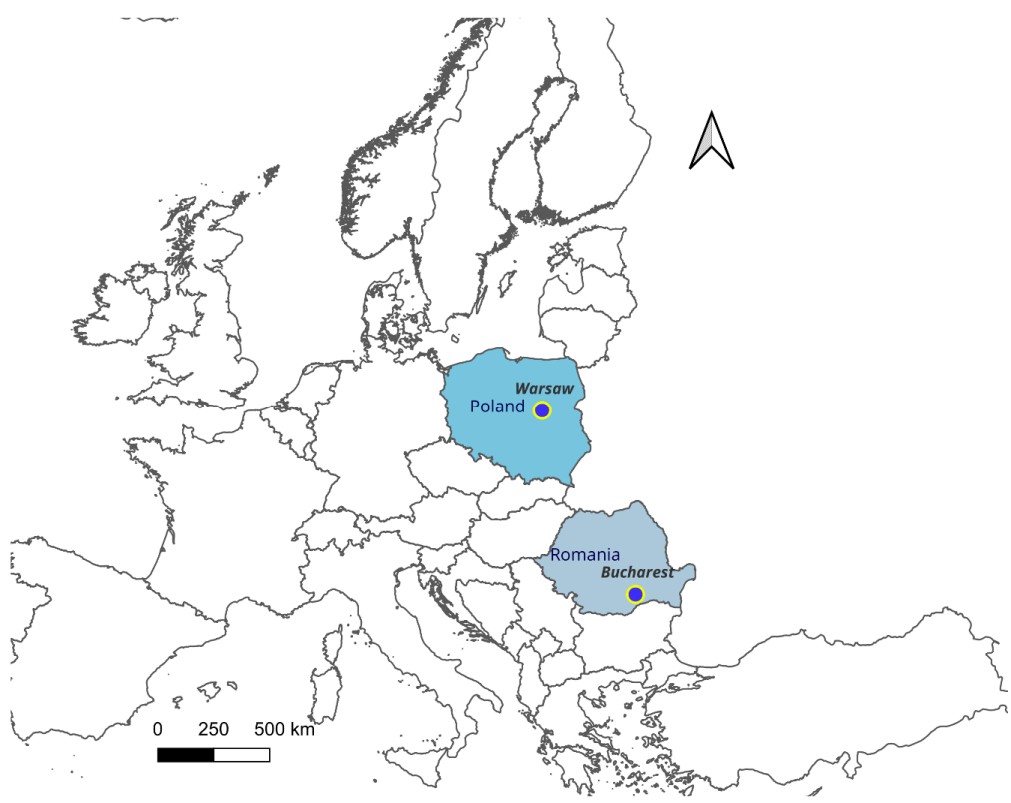

**Figure 2.** Position of Romania and Poland in a European geographical context. The image shows the capital cities where the case studies are located.

The position of the two Romanian case studies is shown in Figure 3; the image also shows the land use. Vacaresti Nature Park (VNP) was selected as the only natural protected area in Romania completely included in an urban area, making it an interesting case study. Băneasa Forest is the only urban forest in Bucharest; it does not currently have protection status as a natural protected area, but is afforded general protection measures imposed by forestry legislation. For example, the forestry code states that an essential principle governing the sustainable use of forests is the promotion and protection of their role, in fact their ecosystem services, and distinguishes categories of forests by role: protection only, and protection with protection [94]. However, a recent project ("People and trees. Management solutions for sustainable development and resilience of Băneasa forest", CIVIS Open Lab Project funded by the University of Bucharest and carried out in 2021 by a consortium composed of the Vacaresti Nature Park Association, Association for Biodiversity Conservation, University of Bucharest, and Ion Mincu University of Architecture and Urbanism) reported that, based on the response from stakeholders, protection status is needed to keep the Băneasa Forest out of reach of real estate developers.

In Poland, the Kabacki and Bielanski Forests were selected as case studies. Both areas lie within the city of Warsaw: Bielanski Forest is located in the north-west and Kabacki Forest is located in the south (see Figure 4, also indicating land use).

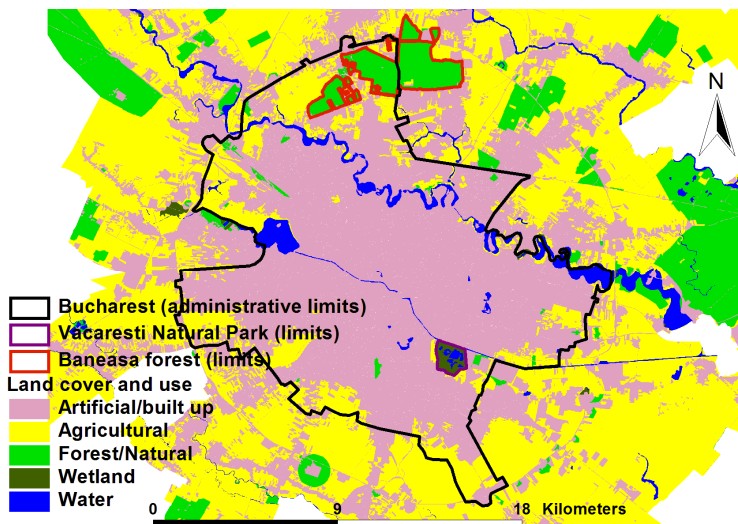

**Figure 3.** Position of the two Romanian case studies in Bucharest (administrative limits shown with a black a contour) in the land use context. The limits of the Vacaresti Nature Park are displayed using a purple contour (bottom part of image, close to the center), and those of the Băneasa Forest are displayed with a red contour (top center).

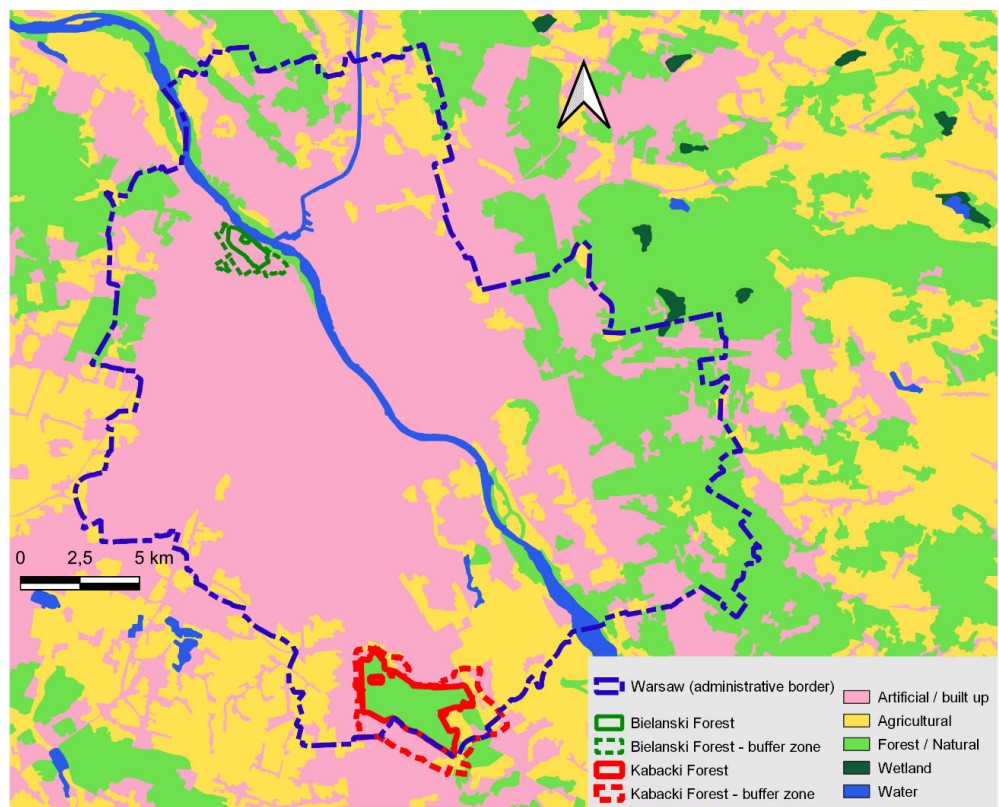

**Figure 4.** Bielanski and Kabacki Forests in Warsaw in the land use context.

Both forest complexes are covered by forms of nature protection and are landscape reserves; the Bielanski Forest since 1973 and the Kabacki Forest since 1980. Both are forested areas (at least partially) located on the Warsaw escarpment (Vistula escarpment), the most characteristic natural element of Warsaw's landforms, and are also protected, shaping the landscape and spatial structure of the city. Both reserves are an important element of the natural system of the city of Warsaw [95]. Both forest complexes perform important social functions (including recreational). They are nature reserves open to the public and it

is possible to walk along designated trails and cycle along designated routes within the reserves.

In addition, both forest complexes contribute positively to the attractiveness of living in the Bielany and Ursynów districts. This can be seen by comparing the prices of flats in Warsaw (both districts are among the most expensive). Both districts are subject to very strong development pressure, especially in the immediate vicinity of the analyzed forest complexes.

## 3. Results

### *3.1. Comparison of Nature Conservation Systems and Urban Pressures in Romania and Poland*

3.1.1. Romania Nature Conservation

(a)    Historical background

In 1782, a law intended to protect the forests of Bucovina (territory becoming later part of Romania as we know it today), introduced the idea of what is known today as 'sustainable development' and the importance of protecting natural elements. This law specified the importance of forest protection so that next generations could also benefit from it [74]. Romania declared its first official natural protected area in 1932 [96], although some areas with restricted human activities appeared in 1919 [97]. The modern system of natural protected areas appeared during the communist period, using the model and categories recommended by the International Union for the Conservation of Nature (IUCN) [98]. By 1981, 260 natural protected areas were declared [99]. However, protection status was not always enforced [100], and the protection process was merely internal, without being connected to international programs designating networks of protected areas. The process of declaring natural protected areas did not continue immediately in the post-communist period, but peaked around 2005-2007. This time period is related to the accession of Romania to the European Union, which required, among other things, the implementation of Natura 2000 natural protected areas (Areas of Special Conservation Interest and Special Protection Areas) [101]. The process was not easy [102], generating overlap between new and existing categories [103] and a consequent lawsuit by the European Union [104]. Currently, accounting for the overlaps, the 1572 Romanian natural protected areas cover 18% of the national territory [101].

(b)    Description of the current system

1.    Principles

There are several key principles in the Romanian legislation governing the declaration and management of natural protected areas. First, conservation is not perceived as strict preservation of species and habitats, but maintenance of ecosystems within the limits of their carrying capacity. According to the second principle, the declaration of natural protected areas should ensure that they are representative of national bio-geographical diversity. Previous studies have shown that the spatial coverage of natural protected areas is good with respect to priority habitats, landform diversity, and key ecosystems [101]. However, other studies indicated that the overlapping of different categories has generated many environmental issues and is likely to diminish protective efficiency [103]. Third, management relies on inner zoning, enabling the protection of core areas by gradually permitting human activities around it, based on their impact. In more detail, core areas are included in the 1st category of IUCN (strict nature reserves) and surrounded by buffer areas and sustainable development areas (the latest corresponding to IUCN VI). The other principles refer to public participation of the local population in drafting the management plans of protected areas, and international cooperation in the process of declaring and managing natural protected areas.

2.    Categories

The Romanian protected areas (regardless of being protected as natural or cultural heritage) include, from a planning perspective:

(1) The section of the National Spatial Plan dealing with protected areas. The National Spatial Plan is the planning document that deals with national territory; it consists of specialized sections, each one approved by law [64]. Section 3, "Protected areas", was approved in 2000. Although many new natural protected areas were declared after 2005, as part of the accession of Romania to the European Union, the plan was not updated accordingly. The plan covers both natural and cultural heritage.

(2) The system of natural protected areas, as part of the environmental legislation. It includes areas of local interest, established at the county level, and those of natural interest, including areas declared based on the European Union requirements, those based on different international conventions and programs, and those designed based on the national legislation, devised in accordance with the guidelines of the International Union for the Conservation of Nature.

(3) Historical monuments, managed by the Ministry of culture, are also declared locally and nationally. There are four categories defined based on their nature: (a) archaeological monuments, (b) architectural monuments and assemblies, (c) public forum, and (d) memorial or funerary monuments.

(4) The national regulation of planning (providing general provisions for all plans) requires the establishment of "protection areas" that prevent the construction of any buildings around different features, including monuments, waters, forests etc. Basically, the law establishes the size of buffers for each feature.

(5) The forestry law has provisions for the protection of some types of forests, regardless of their being included into natural protected areas or not. The first class includes forests with special protection functions, such as those along water courses, those protecting special types of soils, those mitigating climate change, those with a social role—such as park forests or those alongside roads, those with scientific value, and those with outstanding biodiversity. The other classes include forests with a productive role in addition to their protective role. There are management recommendations for each category.

The Romanian system of natural protected areas includes areas of national interest (categories devised in accordance with the IUCN guidelines), areas of international interest (devised based on international conventions or programs), areas of European interest (categories belonging to Natura 2000 network), and areas of local interest. There are slight differences from the IUCN designations. Natural monuments are, according to IUCN, "areas." In Romania, they include species and even individuals (such as old trees) and their management includes delimiting an area around the individual or core habitat, so there is a similarity between them through the management practices. Moreover, the efficiency of the overall protection system is contested in the literature [105,106]. Table 1 presents the Romanian categories of natural protected areas and their IUCN and other corresponding categories. The table indicates that all Romanian categories of protected areas correspond to the IUCN or international categories.

**Table 1.** Romania's system of natural protected areas in the context of IUCN and other international guidelines.

| Romania | IUCN | Others |
|---|---|---|
| Scientific reserve | Ia: Strict Nature Reserve Ib: Wilderness Area | |
| National park | II: National Park | |
| Natural monument | III: Natural Monument or Feature | |
| Natural reserve | IV: Habitat/Species Management Area | |
| Natural park | V: Protected Landscape/Seascape | |

**Table 1.** *Cont.*

| Romania | IUCN | Others |
|---|---|---|
| No category, although areas with such designation are found within large natural protected areas (reserve of biosphere, park) | VI: Protected area with sustainable use of natural resources | |
| Reserve of biosphere | | "Man and Biosphere" Program |
| Wetlands of international importance | | Ramsar Convention |
| Natural world heritage sites | | UNESCO |
| Site of community importance | | Natura 2000-Habitat |
| Special protection area | | Natura 2000-Birds |
| Geoparks | | European Geoparks Network, Global Geoparks Network |

Source: Produced by the authors.

3.  Management %endenumerate

For most areas, management is performed based on the recommendations of IUCN or of the program or convention involved in their creation. To date, it generally involves the existence of a custodian (legal entity, e.g., state company, national society, local authority, decentralized structure of central authority, education or research institution, association for intercommunity development, museum) and a management plan. A notable exception is the Danube Delta Biosphere Reserve, which has its own administration and a special law [107]. However, not all Romanian natural protected areas benefited from custodians and/or management plans [101], and as a result, they were affected by anthropogenic impacts [107]. There are also inherent conflicts between the restrictions imposed by the protected area status and lack of compensatory measures for the local population [107].

Moreover, a new National Agency of Natural Protected Areas was created in 2016 with deleterious effects. Whereas several people, usually dedicated, used to manage a single area before, the Agency now has only 2-3 people dealing with all of the natural protected areas from each of the 41 Romanian counties [108]. As a result, the declaration of a natural protected area is unable to stop changes in land cover and use even within the area's limits. Phenomena such as deforestation and urbanization still occur, illegally or by derogatory practices. The situation can be changed by local agreements. For example, the only natural park of Bucharest is still managed by the previously responsible NGO based on an agreement with the municipality, but there are no funds available for new activities [108].

4.  Urban pressure

Studies looking at land cover and use changes in Romania based on data covering the period 1990–2018, coinciding with liberalization of the economy, have indicated that urbanization is a major driver of change [109]. In addition, urbanization due to tourism affects areas naturally attractive for tourists, such as mountain and coastal areas. The pressure is also felt within urban areas, where the green infrastructure is fragmented, lost, and even eliminated in an aggressive morpho-typological urban structure through continuous and brutal expansion, which does not take into account the historical and evolutionary morpho-structural seeds of the urban tissue of specific and particular cultural landscapes, or the 'genius loci' of areas. The main driver is, ultimately, property restitution [109]. After 1989, forests and agricultural land previously belonging to the state, which confiscated them when the communist regime came into place, were returned to their owners, in fact to their successors, who saw opportunities to make money. Indeed, many restituted forests were cut off [110] and many restituted agricultural lands were abandoned [111], as indicated

by land cover analyses pinpointing deforestation and urbanization as dominant changes in the post-communist period [109]. Land abandonment (including agricultural land) can be seen as a precursor of its transformation into constructions [112], with a significant negative contribution to the transformation (sometimes irreversible) of existing natural, anthropic, and cultural landscapes. Although restituted forests were not cut off within the city limits [113], the pressure existed, as indicated by the analysis of Băneasa forest (see the results).

The effects of this restitution process were aggravated by the fact that the central and local administrations had little or no land reserves for new developments; thus, the solution was to convert natural and agricultural land, by means of planning. In more detail, the territory of each administrative unit includes land that can be built up and land assigned to other uses (natural or agricultural), including parts of natural protected areas situated within the administrative territory of cities. Spatial plans provide for taking land from the second category and including it in the first, as a precursor of urban sprawl. While this process can take a long while, as it requires first changing the plans for the larger area (county, city etc.) [114], it is facilitated by the practice of "derogatory planning" [115], consisting of over developments exempted from planning regulations due to corruption or political influences. More precisely, developers deviate from the planning requirements and then apply for an exemption, which can be obtained, in the aforementioned circumstances, faster than the normal process would take.

### 3.1.2. Poland
### Nature Conservation

(a)    Historical background

In Poland, the history of nature protection dates back to the Middle Ages, when restrictions on hunting aurochs were introduced or the felling of particularly impressive trees, such as oaks, was prohibited. However, the system of nature protection as we know it today began to take shape in the 19th century, when the first nature reserves were established and numerous animal and plant species were put under protection. Even before World War II, Poland had a law for nature protection (1934), and scientists pointed out the necessity of landscape protection as a whole, including both the natural and cultural environment. The first national parks were established in Poland in 1932, including Białowieski National Park and Pieniński National Park. Further development of national parks took place from the 1950s to the 1970s. By 1974, a total of 13 national parks had been established in Poland. In subsequent years after political and economic changes in the 1990s, further national parks were established, the youngest being the Warta Mouth Park established in 2001. In the last 20 years, no new national park has been established, despite the efforts of naturalists. Generally, the system of nature protection in Poland was and is shaped in accordance with international guidelines resulting from the recommendations of the International Union for Nature Conservation (IUCN) and the Convention on Biological Diversity [116]. Poland's accession to the European Union in 2004 was connected with the implementation of the Natura 2000 network of protected areas (SACs and SPAs) [117]. This was the only change in terms of protected areas in the post-communist period. It is worth noting that individual forms of nature protection are not separable—one area may have different protection categories. In Poland, 37.9% of the total area is covered by forms of nature conservation according to the World Bank [118]. However, these are very diverse and sometimes weak forms of nature conservation. According to the Central Statistical Office, 32.3% of the country's area was covered by forms of nature conservation at the end of 2019, but this indicator shows very high regional variation—the largest area covered by nature conservation occurs in Świętokrzyskie Province (64.9%) and the smallest area is in Dolnośląskie Province (18.6%). In Poland, there are 1501 nature reserves covering 0.5% of the country's area and 125 landscape parks covering 8.4% of the country's area. The largest number and area of protected landscape areas in Poland is 387, covering 28.5% of the country's area. The Natura 2000 network established in 2004 comprises a total of 849

SACs (11.2% of the country's area) and 145 SPAs (15.7% of the country's area). Natura 2000 network areas mostly overlap with other protected areas, e.g., national parks or nature reserves, thus they do not significantly affect the area under protection [119].

(b)　Description of the current system

　　1.　Principles

The principles and objectives of nature conservation in Poland are laid down in the Act on Nature Conservation. This Act stipulates that nature protection consist of the conservation, sustainable use, and restoration of natural resources, formations, and components, including urban and rural greenery and trees. The objectives of nature protection declared in the Act include the maintenance of ecological processes, stability of systems, maintenance of biodiversity, and also protection of landscape values, greenery in towns and villages, and afforestation. The shaping of appropriate human attitudes is also mentioned as an objective.

　　2.　Categories

In the Polish spatial planning system, issues concerning protected areas are addressed at different levels and in different ways:

(1) As it stands, Poland does not have a national act on spatial planning, which is highly criticized [120]. Based on statutory solutions, the Concept of National Spatial Planning is no longer valid, and the document that would replace it—the Concept of National Development—is still under development.

(2) The system of natural protected areas, as in Romania, reflects the requirements of the European Union, international conventions, as well as programs and guidelines of the International Union for Conservation of Nature. Protected areas are established at national, regional, and local levels, depending on the type of area. The detailed scope of nature protection forms is defined in the Nature Protection Act. 3. The effectiveness of the nature conservation system in Poland is undermined by the overlapping (in some areas) of various forms of protection. In addition, local self-governments have significant powers regarding opinions for the created forms of nature protection. This often leads to blocking the creation of new or extending existing forms of nature conservation [87,116].

(3) Monuments are defined in the Act for the protection and care of monuments. The Act defines the principles of creating the national program of protection and care over monuments. It also introduces property restrictions in this respect. A movable monument, an immovable monument, and an archaeological monument are distinguished.

In Poland, there is no uniform framework related to restrictions on development for conservation areas. They are adapted to specific categories. Thus, in the case of forms of nature protection, buffer zones are designated depending on the specific form. Moreover, the very possibility of a negative impact of an investment on a natural area under a form of nature protection constitutes grounds for blocking the investment. In the case of monuments, there is the so-called "view protection". On the basis of this protection, buildings which could adversely affect the perception of cultural values of a given object are restricted. However, in both cases, this is very subjective and undefined.

(4) Elements concerning protected areas are also included in the Environmental Protection Act. This applies to particular parts of spatial development plans—the principles of environmental and nature protection and the principles of protection of historical monuments and cultural heritage.

(5) In the Polish system, forms of nature protection are distinguished as national parks, nature reserves, landscape parks, areas of protected landscape, Natura 2000 areas, nature monuments, documentation sites, ecological sites, nature and landscape complexes, and species protection of plants, animals, and fungi. The different forms of nature protection refer to different categories of land.

As with the analysis of Romania's forms of nature conservation, Table 2 summarizes the forms of nature conservation in relation to IUCN categories and guidelines or other international conventions. The table indicates that all Polish categories of protected areas

correspond to the IUCN or international ones. It is worth noting that IUCN categories do not have a formal assignment to legal forms of nature conservation in Poland, hence the difficulty in classifying some of them.

**Table 2.** Poland's system of natural protected areas in the context of IUCN guidelines.

| Poland | IUCN | Others |
|---|---|---|
| Nature reserves (only strict reserves and strict protection areas in national parks) | Ia: Strict Nature Reserve<br>Ib: Wilderness Area | |
| National parks (although the precise definition according to IUNC is met by 15 of the Polish national parks and the rest may be included in category V) | II: National Park | |
| In principle does not occur in Poland within the meaning of IUCN regulations | III: Natural Monument or Feature | |
| Nature reserves (except strictly classified in category I) | IV: Habitat/Species Management Area | |
| Landscape parks | V: Protected Landscape/Seascape | |
| Protected landscape areas | VI: Protected area with sustainable use of natural resources | |
| UNESCO Reserve of biosphere | | "Man and Biosphere" Program—in Poland there are 11 sites |
| Wetlands of international importance | | Ramsar Convention—in Poland these are 19 areas |
| Natural world heritage sites—the World Heritage List | | UNESCO—16 objects are listed, including 2 natural areas. |
| Site of community importance | | Natura 2000—Habitat |
| Special protection area | | Natura 2000—Birds |
| Geoparks | | European Geoparks Network, Global Geoparks Network—in Poland these are 3 sites |

Source: Produced by the authors.

3. Management

The management system of protected areas depends on their legal status. For national parks, nature reserves, landscape parks, and Natura 2000 areas, there is an obligation to develop a protection plan. Additionally, protection plans are developed for UNESCO World Heritage sites, biosphere reserves, and Ramsar sites. The authorities in charge of managing protected areas in Poland include a minister responsible for the environment, general director for environmental protection, voivode, regional director for environmental protection, marshal of the voivodeship, director of the national park, starosta, head of the commune, and mayor or president of the city. It is worth noting that the overlapping of different forms of nature conservation poses challenges to the management of the area and its effective protection. For example, the Białowieża Forest has as many as 7 overlapping statutory forms of nature conservation and two international ones (biosphere reserve and UNESCO World Heritage site) [121].

In Poland, local governments also have significant power in the field of nature protection management. They consult on the establishment, liquidation, and change of borders of national parks, landscape parks, and protected landscape areas. In addition, communes can create or liquidate natural monuments, documentary sites, ecological grounds, and natural landscape complexes. In the context of linking nature protection with cultural landscape protection, the ability to establish a cultural park (which is a form of monument protection) should be considered important—there are 37 cultural parks in Poland.

In the case of Poland, spatial planning instruments have a very serious impact on actual nature protection, especially in areas that are not territorial forms of nature protection. In particular, local spatial development plans adopted at the commune level may play an important role here since they are generally binding legal acts. However, their adoption for a given area is not obligatory and depends on the free decision of the municipal authorities. Nevertheless, if a spatial development plan is adopted, its obligations include the principles of environmental and nature protection. It is on the basis of these principles that it is potentially possible to introduce environmentally justified restrictions on land development [122]. For such a restriction to be possible, it must comply with the constitutional principle of proportionality [123]. Planning interference is equated with restricting private property rights. For such a restriction to be possible, it must be strongly justified but this sometimes poses a problem [124]. For this reason, many cities do not have enacted zoning plans, or the zoning plans do not protect the natural values of areas. Thus, for example, while national parks or nature reserves are sufficiently protected, there is no uniform legal and public framework providing a basis for nature conservation in cities [125,126].

4.  Urban pressure

Urban pressure in Poland is very high. It remains impossible to contain it using spatial policy instruments. This is related to flaws in the construction of individual legal solutions. Strategic planning acts at the local level, i.e., studies of spatial development conditions and directions, should define local spatial policies [127]. Even if in practice they actually attempt to do so, they are non-binding and very often not taken into account in any way. As indicated above, local spatial development plans can theoretically introduce binding restrictions on development. In practice, however, areas subject to investment pressure are often not included in plans [128]. In another variant, the plans adopted for such areas do not contain bolder provisions that protect nature. One reason is the financial consequences of enacting plans [129]. To enact a plan that reduces the value of someone's property, the municipality must pay high compensation. In a situation where a spatial development plan is not enforced, an investment in a given area is carried out on the basis of an even worse solution—a decision on development conditions [130]. These are administrative decisions concerning individual applications. Thus, nature protection cases (without a strong basis in the form of nature protection) are taken into account to a negligible extent.

In Poland, in-depth analyses of spatial chaos and its costs were prepared by the Committee for Spatial Planning of the Country of the Polish Academy of Sciences [22,131]. Very serious problems with urban pressure were found both in suburban areas and in cities. Development (very often carried out on the basis of decisions on development conditions) aggravated the diagnosed problems, including poor state of technical infrastructure, lack of territorial development, morphological-functional chaos, excessive location of development on agricultural land, oversupply of investment land with low location potential, and low efficiency of settlement.

### 3.2. Comparison of Selected Case Studies in Romania and Poland

#### 3.2.1. Romania

The land use maps are shown in Figure 5 (Băneasa Forest) and Figure 6 (Vacaresti Natural Park), based on 2018 Urban Atlas data. The images indicate that natural/forest ecosystems are the main land use, although agricultural, built-up areas, and (to a negligible extent) other uses cover a small share of the total area.

#### 3.2.2. Poland

The land use map of the Bieleński and Kabaty Forests are shown in Figures 7 and 8, respectively. The images shows that the main land use is forest ecosystems. In both cases, it is worth noting the buffer zone of the reserves and land use in this area.

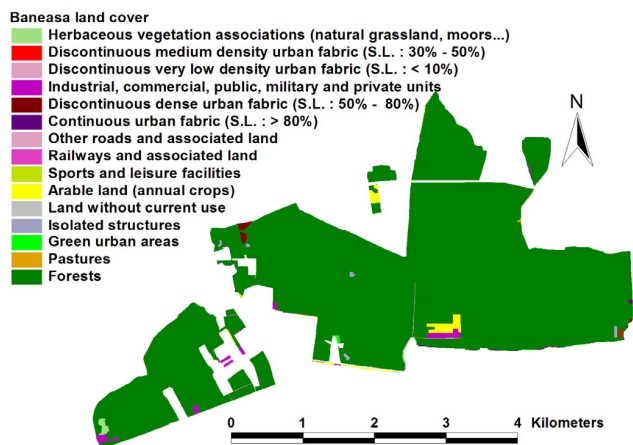

**Figure 5.** Land cover of Băneasa Forest based on 2018 Urban Atlas data. As the name of site suggests, the image shows that the dominant land use is represented by forests.

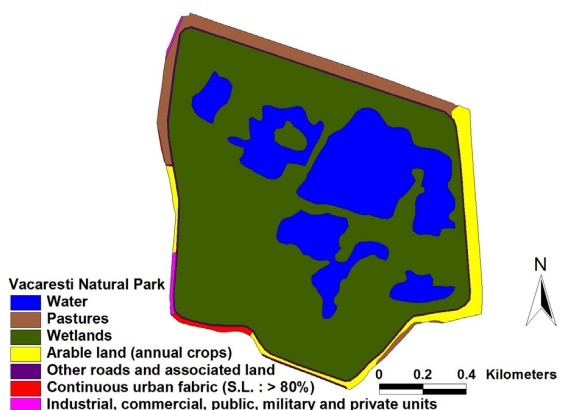

**Figure 6.** Land cover of Vacaresti Natural Park based on 2018 Urban Atlas data. The image shows that most of the area is occupied by water and wetlands.

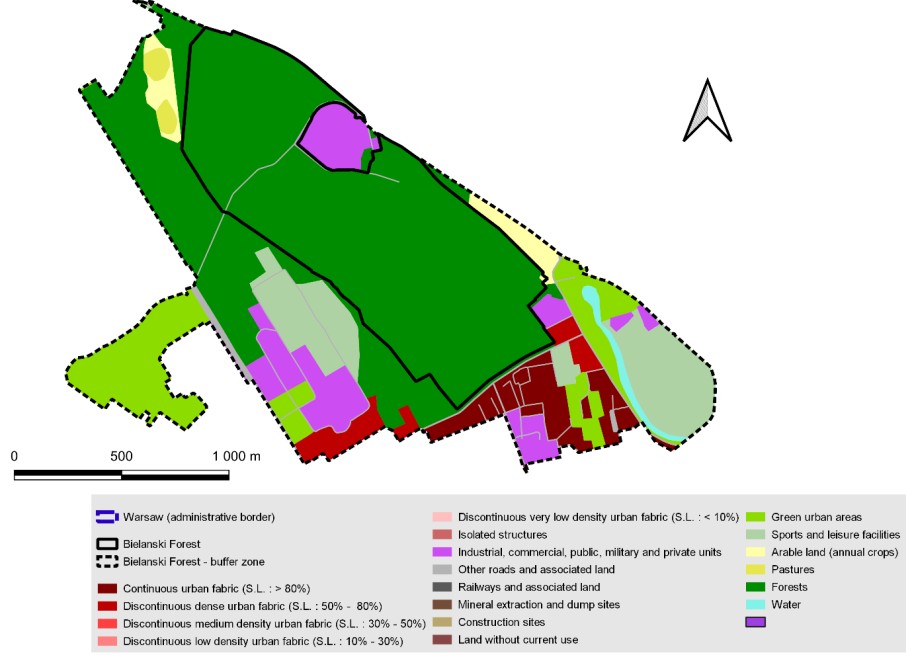

**Figure 7.** Land cover of Bielanski Forest, based on 2018 Urban Atlas data. As the name of site suggests, the image shows that the dominant land use is represented by forests.

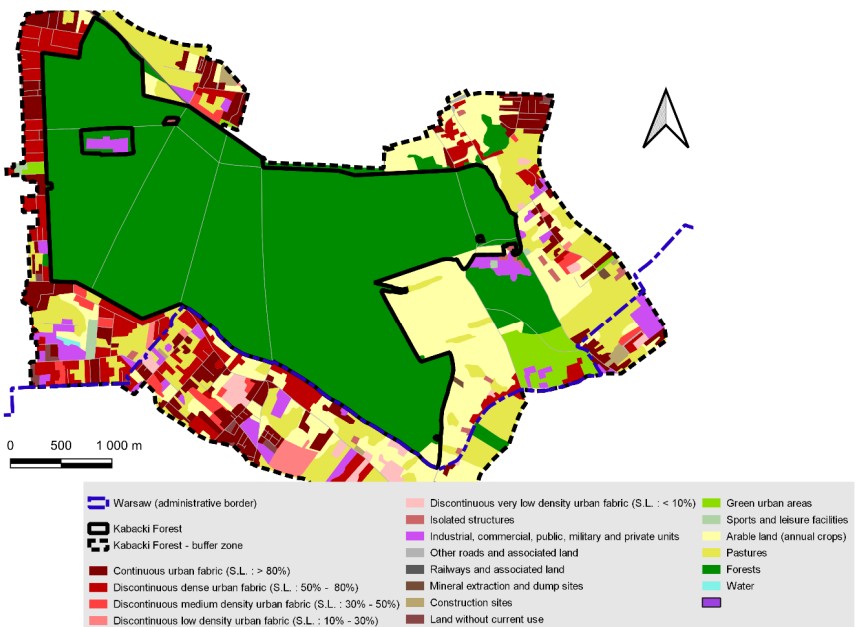

**Figure 8.** Land cover of Kabacki Forest, based on 2018 Urban Atlas data. As the name of site suggests, the image shows that the dominant land use is represented by forests.

3.2.3. Comparison of Case Studies

Basic comparative information about the 4 case studies analyzed is summarized in Table 3. Specific information on each case study follows after the table. The table compares the two Romanian and two Polish case studies based on their administrative and physical characteristics.

Băneasa Forest Analysis of Natural Object and Constraints for Spatial Development

The only urban forest in Bucharest, Băneasa Forest, covers an area of 1340 ha (of which 40% is owned by the state) and is located in the north of the city. The forest has a protection regime—the forest is protected due to its social function (park forest).

Apart of its social profile, Băneasa Forest is a biodiversity sanctuary. The area is home to different habitats represented by elements typical of a forest. The most widespread member of the community is birds, with dozens of species choosing to nest in the forest due to the food resource provided by insects. Although more timid, reptiles and amphibians are also present around temporary ponds. Six species of frogs, two species of newts, four species of lizards, and two species of snakes have been identified.

The most common mammals are hedgehogs, wood mice, partridges, squirrels, and moles. However, wild boar and deer are also reported. Băneasa Forest is home to numerous species of trees specific to the Romanian plain area. Common species include the oak, European elm, maple, or linden. From place to place, non-native species have adapted to the local climate, such as acacia, maple, American maple, and Pennsylvania ash (more to the edges of the forest). Shrubs complete the spaces, including hawthorn, horn, blood, shock, and bat.

The ecosystem services offered by Băneasa Forest are multiple and belong mainly to the category of cultural services. Thus, Băneasa Forest is a recreation space for thousands of nearby inhabitants and visitors from other areas. The name of the forest has a strong meaning for Bucharest's inhabitants, offering identity and a sense of place reminiscent of the former legendary forest (Codrii Vlăsiei) that once covered all of the southern part of Romania. Nature observation and sports activities (cycling, running, alert walking) are also included within the category of cultural ecosystem services, A special role of the forest is that of climate regulation. The presence of the forest balances the circulation of air masses, regulates the temperature, and provides a natural filter for airborne particles.

**Table 3.** Case studies from Romania and Poland—a comparative overview. For additional details, please see the detailed presentations following the table.

| Category Analyzed | Romania | | Poland | |
|---|---|---|---|---|
| | Case Study 1 | Case Study 2 | Case Study 1 | Case Study 2 |
| Name | Parcul Natural Văcărești—The Vacaresti Nature Park | Băneasa Forest | Kabacki Forest (forest park and nature reserve) | Bielanski Forest (nature reserve) |
| Total area | 184 ha | 1340 ha | 903.5 ha and the buffer zone of the reserve has an area of 1076.2 ha | 152 ha of the entire Bielanski Forest area, of which 145 hectares is forest land, 130.35 ha is a nature reserve, and the remaining 21.75 ha is the natural buffer zone of the reserve. |
| Position (within the city, adjacent) (including the area in the city—if applicable, i.e., if not all the area is in the city) | Within the city of Bucharest. South–Eastern. 5 km to the city center. | North–Eastern part of Bucharest. The forest also covers part of Voluntari city close to Bucharest. | Southern part of Warsaw, located in Ursynów district and a small part in Wilanow district. All on the territory of the city of Warsaw. | Entirely located in the city in the Bielany district and partially in the center by the Vistula River. All on the territory of the city of Warsaw. |
| Administration (custodian, etc.) | VNP was administrated by the National Agency for Nature Protected Areas (Ministry of Environment). Since 2022, VNP is administrated by the municipality of Bucharest. | Băneasa Forest is under administration of the National Forest Authority (Romsilva). | Regional Director of Environmental Protection in Warsaw based on the 2016 conservation plan and Warsaw municipality (Warsaw Municipal Forests) | Regional Director of Environmental Protection in Warsaw and Warsaw municipality (Warsaw Municipal Forests) |
| Jurisdiction (local, national etc.) | National | Regional and Local | Regional and Local | Regional and Local |
| Status (protected, local restrictions etc.) | Nature protected area, nature park—V IUCN. The area has also a hydrologic protection status: water retention polder. Restrictions associated with nature protection status: no investments other than those supporting nature conservation, nature reconstruction, education, information, and ecotourism. | Forest. Part of the national forest heritage. Protected for its social function. | V forms of nature conservation in IUCN categories. Kabacki Forest is the largest dense forest complex on the left bank of Warsaw. It is also the largest reserve in Mazovia Province. | V forms of nature conservation in IUCN categories. Natura 2000—Habitat (Las Bielanski PLH 140041) Bielanski Forest is one of the most valuable elements of Warsaw's natural and cultural heritage. |
| Dominant land cover | Wetland, open water | Forest | Forest | Forest, river escarpment |

**Table 3.** *Cont.*

| Category Analyzed | Romania | | Poland | |
|---|---|---|---|---|
| | Case Study 1 | Case Study 2 | Case Study 1 | Case Study 2 |
| Type of ecosystem (wetland, forest etc.) | Wetland, urban park | Forest | Forest | Forest |
| Ecosystem services | The area is a compact green space (wetland) easily accessible by the public transportation network. It is also very close to apartment buildings and households, approx. 5000–10,000 inhabitants live around the area. From this point of view, VNP has good ecosystem service delivery potential, especially cultural ES: recreation, cognitive development, sport (running, biking), wildlife, bird watching. The shape of the area (water retention polder) increases the possibility of developing nature-based solution for water flow regulation and runoff mitigation. | The ES produced by the forest are multiple and belong mainly to the category of cultural services (recreation, education, aesthetics, sense of place). A special role of the forest is that of climate regulation. The presence of the forest balances the circulation of air masses, regulates the temperature, and provides a natural filter for airborne particles. In the category of production services: wood, medicinal plants, berries, mushrooms. | Ecosystem services—referring to the various types of goods and services that benefit humans and contribute to human prosperity. Regulating ecosystem services, Kabacki Forest involves the preservation of habitats and species (maintaining conditions in which species can live, feed, and reproduce) and also benefits related to ensuring air quality and influencing the climate. This category also includes benefits related to soil protection and purification (such as influencing the nutrient cycles, accumulating organic material, or preventing erosion) and water resource protection and purification (water filtration and retention at various scales). The forest also has social functions. | Bielanski Forest provides ecosystem services like any forest complex, such as Kabaty Forest. Its high natural value is determined by old tree stands with a natural primeval structure and preserved richness of flora and fauna with the presence of species unique in the city and region. As a result, Bielanski Forest plays an important role in preserving the biodiversity and ecological corridors of Warsaw. Among other things, it is the only refuge of a number of plant and animal species within a radius of several kilometers. |

Source: Own Studies.

Over time, the surface area of the Băneasa Forest has decreased due to the reconstitution of property rights after 1989 and the forest is subject to permanent pressures determined by improper urban planning and chaotic expansion of the city (urban sprawl). As in the case of other green spaces, these pressures are mainly of a real estate nature and concern the construction of new residential complexes. The property regime is mixed, with 65% represented by private areas and 35% represented by public property of the state. In 2020, the state-owned forest (400 ha) received a protection regime due to its social profile. However, the efficiency of the protection is arguable. Figure 9 is based on data freely and unrestrictedly available to everyone from the National Office of Cadastre and Real Estate. The map shows that the forest is not a continuum, but distinguishes property limits within it, based on different ownerships. This is an indication that the forest can be divided into smaller parcels and owners can change the land use if the overall protection ceases.

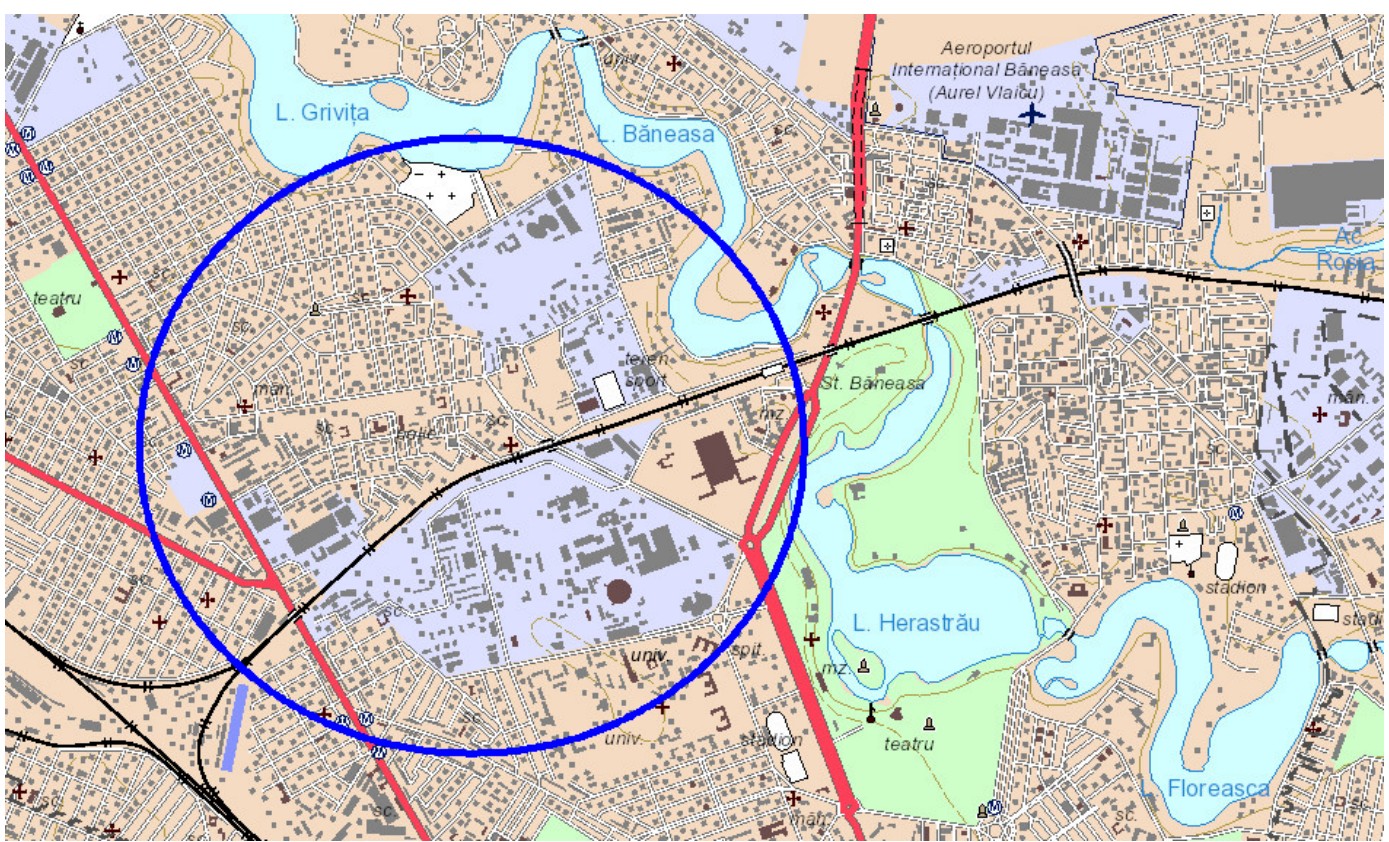

**Figure 9.** Property in Băneasa Forest (purple area within the blue circle) based on data from the Office of Cadastre and Real Estate. The map displays property limits, based on different ownerships, showing that the forest can be divided into smaller parcels and owners can change the land use if protection ceases.

The public property part of the forest owned by the state came under protection in 2020 due to its social functions (park forest). According to the law, landscaping work (alleys made of ecological materials with a maximum width of 2.0 m, benches, lighting, ecological toilets, visiting points, and wooden constructions with a maximum surface of 15 m$^2$) must be carried out without extracting the trees. Afforestation can be performed with species that are not of the fundamental natural type, including exotic species, instead of extracting trees.

Băneasa Forest is part of the national forest heritage, according to the Romanian Forestry Code, and is thus subject to a series of restrictions from the urban point of view. The official status of the forest is the main justification for planning restrictions in the area. The forest does not have special status within the city, so it requires different planning. It

is not recognized as such by the city's development plans. There is no urban connection between the city and the forest. This is the reason why civil constructions (residential neighborhoods) have appeared in the immediate vicinity of the forest but also inside it in the last 20 years and there is a permanent pressure in this regard.

Out of the total of 1340 ha of forest within the Băneasa Forest, 63% is privately owned. Real estate pressure causes the forest to be divided into smaller and smaller parcels with each owner having the legal right to build a holiday home/cabin that permanently removes an area of 200 m$^2$, but not more than 5% of the surface of the property, from the forest heritage, according to article no. 37 of the Forestry Code. Through the subdivision of the forest, permanent deforestation, and the construction of houses, the forest is fragmented and the social function disappears by restricting access.

Forestry regulations applied by local forestry departments produce an internal zoning of forests, with areas enforcing more or less stricter protection measures and imposing different restrictions for each type of zone. However, the data on the inner zoning are not publicly available.

Specific forestry restrictions differ from the rest of the city, as the internal zoning is made in accordance with forestry legislation and not with the regular planning spatial regulation. From the spatial planning perspective, forested areas are placed in the "non aedificandi" category, meaning the development of constructions is prohibited.

Vacaresti Nature Park Analysis of Natural Object and Constraints for Spatial Development

The Vacaresti Nature Park (VNP) is a wetland and the only nature-protected area located entirely in the urban part of Romania. It is also the biggest green space of Bucharest. It was developed on a former communist construction site (water retention polder) abandoned after 1989. It was established as a nature-protected area in 2016 as a result of a national civic campaign, which was supported by non-governmental organizations, mass media, and research institutions [131], acutely raising the question of opportunity and risks but also ethical values related to the establishment and integration of a such sensitive natural area in a landscape difficult to urbanize [132]. The area is characterized by rich biodiversity reflected mainly by the birds, reptiles, amphibians, and luxuriant landscape and vegetation [133]. In Bucharest, the surface of green spaces has decreased since 1989 due to chaotic planning [134]. The establishment of the VNP in 2016 meant an increase in green space by almost 1 m$^2$ per inhabitant.

The close proximity to urban neighborhoods permits a good flow of ecosystem services, especially in the category of cultural values (education, recreation, scientific research, bird watching, sports). The specificity of wetland and concave shape due to the planned construction of a large lake could support the development of nature-based solutions, especially for rainwater retention and regulations.

According to the Government Emergency Decision no. 57/2007 on the regime of protected natural areas, conservation of natural habitats, wild flora and fauna, a nature-protected area must be established on the basis of scientific evidence (species and habitats). The scientific arguments are analyzed by a group of experts (Romanian Academy) who elaborate an official decision. In Romania, natural protection status covers any form of property, except for situations related to security, public health, or defense. In the case of VNP, the decision was made on a scientific basis: the area provides a rich biodiversity represented mainly by birds (water birds), reptiles, and amphibians, many of which are on the list of national and European protected species.

In addition to its natural protection status, the VNP overlaps with a hydrological protected area. The history of the area has led to this situation. The VNP covers the site of a former construction site (water retention polder). After 1989, the site remained unfinished, but the area appears in official documents as a lake from a legal point of view, so it is subject to the hydrological regime regulated by the Water Law 107/1996, which requires: (1) identification of the legal bases for the restrictions and which public authorities have imposed the restrictions; (2) official justification for the introduction of restrictions in the

study area; and (3) additional (if any) justifications for implementing planning restrictions in the study area.

According to the Emergency Decision no. 57/2007, the nature-protected status includes a series of planning restrictions. Thus, the VNP Management Plan establishes three areas of protection, designated by the Scientific Council of the VNP and proposed according to the natural value of the VNP: (1) "strictly protected" zone in which investments not related to the conservation and protection of nature are prohibited. In this area, activities are allowed that support conservation, education, science, firefighting, and ecological reconstruction; (2) "buffer zone" (sustainable management area). Here, the allowed activities include research, information, grassland maintenance (mowing, exploitation), reconstruction, and ecotourism activities that do not require constructions; and (3) "sustainable development area" in which investment/development activities are allowed, with priority given to those of tourist interest, but respecting the principle of sustainable use of natural resources and prevention of any significant negative effects on the park's biodiversity. According to the law, the development plans of the city (general and local plans) must take into account and adapt to the specifics of the protected area.

The planning restrictions within the VNP are different from the provisions of Bucharest's General Urban Plan (Master Plan) and have a pre-emption in relation to them. Moreover, the provisions of the Government Emergency Decision no. 57/2007 stipulate that urban plans must be adapted to the specific interests of the nature-protected area. Thus, the objectives of conservation, protection, reconstruction, education, and ecotourism are the only ones that will be allowed in the area. The planning restrictions overlap (and are in synergy) with the restrictions imposed by the status of hydro-technical investment. The restrictions imposed by the protection status are meant to protect the natural value of the area and are a good solution to protect nature and inspire similar projects in Bucharest, where green spaces collapse and are fragmented.

Nevertheless, the VNP has been and continues to be subject to obvious pressures from multiple categories, e.g., poaching, arson, illegal waste storage, and even real estate investment attempts. It should be noted as a local peculiarity that although the VNP is located in a seismic zone, the major pressure is due to negative actions against the natural, anthropic, and cultural landscape, with interlinked social and community effects. In more detail, the area is a fragile cultural landscape, not seen as an asset by the neighboring communities because of its history, in particular, interventions in the area during the communist period [135].

Kabacki Forest Analysis of Natural Object and Constraints for Spatial Development

The name of the forest and the reserve comes from the village of Kabaty, which existed on this site in the early 20th century. The terrain of the reserve is flat, but the exception is the eastern part of the forest where there is a fragment of the Warsaw Escarpment. The Powsin Culture Park is located in the southeastern part of the forest. This facility is a very popular place for recreation and leisure, which increases pedestrian and bicycle traffic within the boundaries of the Kabacki Forest reserve. Originally it was privately owned, but the forest was sold to the Municipal Authority of the City of Warsaw (the area purchased was 914 ha) just before World War II. Since that time, i.e., since 1939, the Kabacki Forest has been owned by the city.

During the Second World War, the forest suffered from the effects of warfare. It was also a place of execution of the civilian population of Warsaw (nowadays these areas have historical and educational value). During the war, a secret military unit called "Wicher" operated in the area of the Kabacki Forest, dealing with the decryption of German military codes, including the famous "Enigma." The most important monument is a forester's lodge from 1890.

After the war, the forest was subjected to strong anthropogenic pressure related to intensive forest management, the development of industry in Warsaw, and agriculture, including the grazing of animals in the forest area. Despite this, old-growth forest (species

such as linden, oak, hornbeam, and maple) up to 160 years old has been preserved in some areas. Protected animals include the hedgehog, weasel, black and middle spotted woodpecker, hawk, sparrow hawk, two species of bats (greater mouse-eared bat and red-legged bat), as well as several species of amphibians (lake frog, grass frog, moor frog, common and great crested newts, and wood newt) and reptiles (blindworm, grass snake, sand lizard and viviparous lizard). In the area of the reserve there is also the presence of lily of the valley, a plant under strict species protection. In addition to natural functions, the forest also has material and non-material functions.

The material ecosystem services are primarily benefits related to the provision of food for humans and animals. This category also includes ecosystem services related to the provision of materials for production and also ensuring jobs.

The non-material ecosystem services include benefits related to cultural and spiritual values, education, tourism, and recreation. Within this group, it is worth mentioning those that involve the enjoyment of communing with wild nature during recreation or sports. These ecosystem services also emphasize the creation of conditions for a high quality of life, understood as human wellbeing in the city. The Kabacki Forest complex stretches 4–5 km in length and 2.5–3 km in width. It is bordered to the south by the Piaseczno and Konstancin-Jeziorna municipalities (dominated by single-family housing), to the north by Ursynow (a housing estate with several tens of thousands of inhabitants), and to the west by Puławska Street. It is subject to development pressure from all sides, although the strongest is from the Ursynow side.

Three hiking trails run through the Kabaty Forest: red (9 km long), green (10 km long), and blue (5 km long). There are also two nature trails running through the Kabacki Forest, along which benches, canopies, and information boards have been set up. You can learn more about the history of the place, its inhabitants, and vegetation from them. Nature trail no. 1 has 12 stops and nature trail no. 2 has 10 stops. Both nature trails are about four kilometers long and each takes about three hours to walk. The Warsaw City Forests Nature and Forestry Education Center is located in the vicinity of the historic forester's lodge.

The popularity of the Kabacki Forest as a place for rest and recreation is influenced by its good accessibility by, among other things, the metro line. A local spatial development plan has been in force in the area under consideration since 2010 [136]. According to the provisions of the plan, the area is zoned for the following uses: residential, services, public roads, and technical infrastructure. The stated objectives of the plan include: (1) allowance of residential and service development while maintaining spatial order, taking into account the ecological and protective conditions of the Kabacki Forest nature reserve; (2) determination of conditions for shaping development, the fulfillment of which will allow achievement of appropriate spatial values and the homogeneous character of the shaped space, with particular emphasis on the area of Puławska Street as exposed in the city landscape; and (3) protection of public interests in communications, engineering, and environmental protection.

The importance of residential and service functions relate to prescribing a minimum percentage of 60–80% for the biologically active area.

Polish spatial development plans specify the importance, from the perspective of the subject under consideration, of the purpose of the land (a general indication of the purpose for which the land is to be used), the principles of land development (including the parameters of development), and the principles of environmental and nature protection. The plan in question sets out land uses for housing, services, public roads, and technical infrastructure (i.e., typical for urban areas). However, already in its first part it declares that its purpose is to allow for the development of housing and services, taking into account the ecological and protective conditions of the Kabacki Forest nature reserve, the inclusion of part of the area as a prominent feature in the city landscape, and the protection of public interests in the field of communications, engineering, and environmental protection. These objectives remain the general point of reference when interpreting the more detailed provisions. With the latter, it is worth highlighting the planning restrictions for areas

with residential and service uses. These consist of: (1) exclusion of new development and extensions to existing garages and outbuildings in most of the area; (2) identification of fences to allow the migration of small fauna; (3) setting a significant minimum biologically active area; and (4) absolute obligation to preserve the existing trees indicated in the plan and introduce vegetation compatible with the habitat types.

It should be emphasized that the Polish spatial planning system provides limited possibilities for more extensive planning interferences, even if justified by environmental and natural reasons, since there is a risk that plans may be challenged before administrative courts. Nevertheless, in the indicated area, the plan contains above-standard guidelines in this respect. Particularly noteworthy is the blocking of new development and adapting it to environmental conditions. It is worth noting the restrictions on trees and plants, in a sense going beyond the formal (statutory) scope of the plan.

Bielanski Forest Analysis of Natural Object and Constraints for Spatial Development

The Bielanski Forest Reserve was established in 1973. It is located in the Bielany district between Marymontska, Wisłostrada, and Podleśna Streets.

Bielanski Forest combines the historic primeval landscape with the modern, metropolitan one. It is a unique, on a European scale, enclave of nature preserved in an urbanized environment. It has multiple functions:

- Climatic: it affects the microclimate of the district;
- Scientific: a valuable research object providing an opportunity to track the response of relict biocoenoses and species to the anthropogenic impacts of the big city;
- Didactic: a place for field classes for students and pupils of Warsaw schools;
- Social: a recreational area, satisfying the need for contact with "real" nature to a much better degree than other green areas of the city;
- Landscape: a characteristic component of Warsaw's Vistula River panorama

This area is protected for natural, scientific, and historical reasons. It is of exceptional importance due to its flora and fauna value as a refuge for animals on their migration routes along the Vistula and between Warsaw and the Kampinos Forest. The Bielanski Forest is, in large part, the only remnant of the former Mazovian Forest, preserving the continuity of its forest complex with 400-year-old oak trees bearing witness to this. A particular value of the reserve is its varied relief with a high escarpment and four clearly marked terraces. This relief is varied by ravines located in the northern part of escarpment.

The reserve takes its name from the whiteness of the habits of the monks (Camaldolese monks) who have lived here since the 12th century. The area has survived for centuries from agricultural and settlement development, as well as from progressive urbanization. From the 19th century, the forest served as a recreational area for the inhabitants of Warsaw (it was incorporated into the administrative boundaries of Warsaw in 1930). The Park of Culture was created here, and the influx of people resting and playing caused a threat to the natural environment. In response to this situation, a nature reserve was created and the Park Kultury closed in 1986. Nowadays, it is possible to use the reserve and walking and cycling trails have been marked out in Bielanski Forest [137].

From the scientific research carried out for many years on the Bielanski Forest reserve, it is clear that it is necessary to limit (to a minimum) any activities that cause disturbance of the ecological balance in the reserve area and deterioration of the environmental conditions. In 2016, a Protection Plan for the Bielanski Forest Reserve was established for a period of 20 years, which also included arrangements for the NATURA 2000 site. The Plan states that one of its objectives is to determine how the reserve's buffer zone will be developed in such a way that it will not adversely affect the reserve and the Natura 2000 site, in particular, the purpose and object of protection, i.e., the maintenance of the reserve as a natural ecosystem preserved in the area of the Warsaw urban agglomeration. Annex no. 8 to the Plan contains a map of arrangements to the study of conditions and directions of spatial development of the capital city of Warsaw, local spatial development plans of the capital city of Warsaw, and the spatial development plan of the Mazowieckie voivodeship, which concern the

elimination or reduction of internal or external threats. The entire area of the reserve's buffer zone is divided into zones A–J, for which guidelines for land use and development have been established. Significantly, a very high percentage of biologically active area (from 95% to 80%) is set for zones A–D. In zone E, this indicator is a minimum of 30%. It is worth stressing that the Protection Plan emphasizes that the following are not considered as biologically active areas: greenery designed on roofs and walls of buildings and above- and below-ground structures, gravel, grit, and openwork surfaces. According to the provisions of the plan, biologically active areas are green areas accompanying development, including trees, shrubs, lawns, surface water bodies, and agricultural crops. Such provisions must be transposed into local plans. Thus, areas in the buffer zone of the reserve are to be used as green areas without the possibility of locating cubature objects, with the exception of technical facilities serving the facility. It appears that the Bielenski Forest within the buffer zone is effectively protected from development pressure. A Catholic church and some university buildings are present on the Bielanski Forest site. From time to time, conflict arises over the expansion or redevelopment of these sites.

*3.3. Comparison of Management Institutions and Key Legal Documents That Coordinate Protection of Natural Assets and Urban Planning*

The analysis revealed the complexity of the relationship between the protection of natural assets and spatial planning. This complexity is due to the existence of multiple institutional actors and legal documents. We attempted to present an overall view in Table 4, summarizing only the main stakeholders. The table reveals the great number of actors and laws governing the two fields, suggesting possible overlaps and conflicts.

**Table 4.** Overview of the main actors and laws governing protection of natural assets and spatial planning in Romanian and Poland.

| | | Romania | | Poland | |
|---|---|---|---|---|---|
| | | **Nature Protection** | **Spatial Planning** | **Nature Protection** | **Spatial Planning** |
| Actors | National | Ministry of the Environment, Water, and Forests National Environmental Guard National Agency for Protected Areas | Ministry of Development, Public Works and Administration | Ministry of Climate and Environment General Directorate for Environmental Protection | Ministry of Development and Technology |
| | Local | Local structures of the environmental guard County forestry guard | Local administrations | Local administrations (in cooperation with the Regional Directorate for Environmental Protection) | Local administrations |
| Laws | National | Law on Environmental Protection Forestry Code Law on Natural Protected Areas | Law on Territorial and Urban Planning Law on the Execution of Construction Works General Urban Planning Regulation | Environmental law Nature Conservation Act | Law on planning and spatial development |
| | Local | Decisions of the local administrations | Local spatial plans Decisions of the local administrations | Decisions of the local administrations | Local spatial plans Decisions of the local administrations |

## 4. Discussion

Green areas in a city have a variety of functions, including the strong support of human mental and physical health. Apart from natural functions, the most important functions of green areas for residents of large cities are social, including recreational. Green

areas, including special forest areas, influence the attractiveness of living areas and are therefore subject to strong development pressure [126]. At the same time, they provide basic support for planning operations regarding urban renewal [138], but also a binder for supporting the development of healthy communities, which ensures a favorable climate for harmonious urban life integrating human beings, ecosystems, public health, quality of life, and maintaining a highly viable pathology of the urban organism. The case studies presented (four areas in Bucharest and Warsaw) indicated the natural, material, and non-material functions performed by these areas. Each function is an important element of the whole natural and cultural system of the city.

Analysis of the national protection frameworks showed how the spatial planning systems of Romania and Poland guarantee the protection of sites with significant natural value located in large cities [32]. Both the specific natural characteristics of the indicated sites and institutional attempts at spatial protection of these sites were contrasted [6,8]. In this way, possible approaches to the protection of the indicated areas in the analyzed systems were distinguished. As indicated above, this issue requires special analysis [51]. It is necessary to eliminate the traditional dualisms within planning theory, i.e., the procedural-substantive distinction and theory-practice gap, thus providing a locally diverse and unique interpretation of planning theory at macro-territorial (national) and mezzo-territorial (sub-national) scales by rejecting the idea that local interpretation of theories and their application can be assumed to be consistent with ideas operating at a higher level [139].

Protected areas were identified at the national scale (Băneasa Forest) and at the regional and local scale (others). The basis for setting up land-use restrictions was reviewed in detail. In the case of the Băneasa Forest, the justification for any restrictions was the official (nationally defined) status of the forest. The problem diagnosed in this case is the lack of translation into spatial planning at a lower level. This results in urban pressures concerning both the forest itself and its bordering areas. Protection implemented centrally is not sufficient [41]. Emerging weaknesses were identified related to the subdivision of plots of land into smaller plots and progressive fragmentation of the forest. The extent of planning protection of the Vacaresti Nature Park is definitely better in this respect. This is mainly due to the coordination of urban spatial policy arrangements with nature conservation requirements. The protection regime for designated areas in Warsaw should be assessed differently. The case studies from Warsaw concern areas around which a large number of inhabitants live, such as a housing estate with several thousand inhabitants next to Las Kabacki. Both reserves are open for recreation, which, especially on weekends, results in several thousand people walking through them every day. The surroundings of the Kabacki and Bielanski Forest areas are very attractive places to live. Despite the existence of buffer zones, development pressure is increasing as a result of the weakness of the spatial planning and nature conservation systems in Poland. Buffer zones of protected areas, as studies show, are only able to prevent development to a limited extent. In these cases, attempts to link urban spatial planning with nature conservation goals can also be found [53,72]. While this is implemented within the framework of sectoral nature conservation plans, it does not fulfill its intended role in urban planning [72]. This is primarily due to the deficiencies of the Polish spatial planning system. The best example of this is the spatial plan for the Kabaty Forest. An above-standard approach to the designated area in the plan is noticeable. The aforementioned provisions appear relatively rarely in the spatial plans of other cities. Despite this evident effort on the part of the municipal authorities, there are a number of problems at the implementation stage. These boil down to the fact that the provisions of the plans (e.g., regarding the protection of trees and the requirement to introduce specific vegetation) remain merely non-binding postulates [140].

Contemporary urban and landscape planning theory places the impact of different theories in a social and political context, including those using collaborative, postmodern, and neo-pragmatic approaches. However, some typologies were considered immune to such changes and interpretations of the broad aspects of the planning process, including

the belief that the normative theory changes over time as a result of progress and the introduction of spatial and temporal variation [141].

The characterization of the consequences of planning approaches to the protection of nature conservation areas, from the perspective of different levels of government, should therefore be regarded as an important contribution contained in this article. It is a mistake to delegate responsibility to the central level. It is also a mistake to delegate responsibility to the municipal (local) level without adequate institutional preparation [56], since the planning efforts of municipal authorities will then not always be reflected in practice. At this point, some recommendations for action can be made that will provide a common response to the problems observed in both systems. These boil down to a broader integration of development planning, but also deeper cooperation between different levels [122]. Environmentally valuable areas are an asset of supra-local value, so priorities related to their protection can be defined at the central level [125]. This should be a first step. However, the detailed scope of protection and translation of the identified priorities should already take place at the municipal level. The basis for this should be the formula of an urban spatial plan [130], which should integrate quasi-/semi-natural, anthropic (urban/architectural), and cultural landscape planning as a complex component [142] in a multi-scalar planning approach [143], from the scale of the micro-landscape to spatial planning at the mezzo scale up to the macro landscape scale through territorial/regional planning, and even for the entire European continent. At the same time, the legal framework should provide very specifically for the possible scope of planning restrictions for natural areas. The general definition of this scope is the task of the central authority. In this way, the central authority would formulate the general objectives and implementation tools [84]. On the other hand, it would be up to the municipal authorities to optimally implement these guidelines [144]. The examples of the two countries studied show that this seemingly simple demand is not systemically implemented.

The detailed comparison of the planning regimes of the indicated differentiated natural areas should be considered as an innovative contribution of this article. It was pointed out that, in addition to the rationale indicated in the scientific discussion to date, poor coordination on the part of central and municipal authorities remains a key problem. This problem was presented with examples of specific provisions, the implementation of which has proven to be ineffective in practice. At the same time, this article makes a very strong case for the special role of natural areas under investment pressure. The discussion of the protection of such areas seems to have a broader dimension, as there are numerous cases of overlap or even legislative vacuum, especially when valuable protected areas located and integrated within the administrative territory of human settlements are subject to double pressure. If the solutions proposed for a given system prove effective in this respect, there is a good chance that analogous action will be equally effective with regard to other areas of value from a spatial perspective.

There are, of course, limitations associated with our research. The most significant limitation is the different systems of land use and environmental law. Although a comparison is possible since both countries share common features, the difference in the type of public authority responsible for a particular section of environmental protection makes a simple (e.g., tabular) comparison of selected issues particularly difficult. A similar limitation is the difference in processes for drafting local spatial plans, including their obligatory nature [64].

Further research directions can also be identified:

- A comparison of the planning protection of environmentally valuable areas in other Central and Eastern European cities. The countries of Central and Eastern Europe are similar in many respects in the sphere of spatial policy. The criterion proposed in this article, involving the balancing of central and urban perspectives through an integrated multi-scalar approach, can be an important point of reference;
- Another aspect of research that should be developed relates to thorough studies of the ecosystem services provided by naturally valuable areas in cities, especially related to

non-material aspects. In addition, environmentally valuable areas in cities should be subject to multifaceted environmental valuation [145,146];

- On a similar note, it is worth reviewing the feasibility of planning protection for areas of value using a trans-disciplinary approach for complex landscapes [147] and a cultural heritage perspective [148], linking national and local environmental and urban planning [138]. This thematic scope also needs to be substantiated in international comparisons that discuss the importance of current landscapes as a development milestone in spatial and territorial planning theories [149], as well as different approaches to the new spatial urbanism or other directions and theories of urban and territorial planning [150];

- Another important topic concerns the further comparison (alignment) of nature conservation-related, planning, and legal terminology. Many systems have major discrepancies in this respect. These are often the basis for the weakness of spatial plans insofar as they relate to the natural sphere. This issue needs to be sorted out more extensively.

## 5. Conclusions

The research presented here analyzes the role played by natural areas. The ecosystem services provided for the residents of Warsaw and Bucharest in the analyzed areas were identified. Although different, Romania and Poland share similarities, including those resulting from accession to the European Union after the dispatch of communist regimes. However, differences between countries were the main limitation and were overcome by employing a multilayered comparative approach. Key barriers related to planning protection were also identified. It turns out that where there is serious urban pressure in the indicated areas, protection through the construction of spatial policy solutions has to be particularly considered in relation to concerted planning operations at a geographical landscape scale. This paper identifies key reasons for the poor planning protection of designated areas, despite the often goodwill of central and municipal authorities. These problems can be put down to a lack of thoughtful integration of development policies and in-depth coordination between public authorities at different levels, which is partially rooted in the post-communist social and ecological transition. This is particularly evident at the confluence of spatial and nature conservation policies. In addition, areas of natural value determine the attractiveness of a place to live. Investment pressure, especially development pressure in the immediate vicinity of environmentally valuable areas, has a negative impact on the functioning of natural areas, requiring complex and concerted urban planning actions in an integrated process. In this context, the role of discussing the concrete (also legal) construction of spatial policy instruments requires special emphasis. Especially in a situation of significant investment pressure, such construction needs to be particularly well thought out in order to have deep and significant implications on quality of life, the vulnerability of natural, anthropic, and cultural landscapes, urban and social life, urban pathology, and human health.

**Author Contributions:** Conceptualization, P.L.-K., M.N. and A.-I.P.; methodology, P.L.-K., M.N. and A.-I.P.; software, P.L.-K., M.N., A.-I.P. and D.B.; validation, P.L.-K., M.N., A.-I.P., D.B., C.C. and A.-I.G.; formal analysis, P.L.-K., M.N., A.-I.P., D.B., C.C. and A.-I.G.; investigation, P.L.-K., M.N., A.-I.P., D.B., C.C. and A.-I.G.; resources, A.-I.P.; data curation, P.L.-K., M.N., A.-I.P., D.B. and A.-I.G.; writing—original draft preparation, P.L.-K., M.N., A.-I.P., D.B., C.C. and A.-I.G.; writing—review and editing, P.L.-K., M.N., A.-I.P., D.B., C.C. and A.-I.G.; visualization, P.L.-K., M.N., A.-I.P., D.B., C.C. and A.-I.G.; supervision, P.L.-K., M.N. and A.-I.P.; project administration, P.L.-K., M.N. and A.-I.P.; funding acquisition, A.-I.P. All authors have read and agreed to the published version of the manuscript.

**Funding:** This research received no external funding.

**Institutional Review Board Statement:** Not applicable.

**Informed Consent Statement:** Not applicable.

**Data Availability Statement:** Not applicable.

**Acknowledgments:** This research presents, partially, results of the CIVIS Open Lab project "People and trees. Management solutions for sustainable development and resilience of Băneasa forest" coordinated by the Faculty of Biology at the University of Bucharest in 2021, the doctoral theses "Natural protected areas from the urban environment" by doctoral student Atena-Ioana Gârjoabă and advisor Cerasella Crăciun at Ion Mincu University of Architecture and Urbanism, and "Assessment of ecosystem services of green-blue infrastructure in urban areas. Case study: Vacaresti Natural Park" by doctoral student Dan Bărbulescu and advisor Geta Rîșnoveanu at the University of Bucharest.

**Conflicts of Interest:** The authors declare no conflict of interest.

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
