# Peer review of "Protection of Environmental and Natural Values of Urban Areas against Investment Pressure: A Case Study of Romania and Poland"

_land, doi:10.3390/land12010245_

Round 1

Reviewer 1 Report

The paper entitled “Protection of Environmental and Natural Values of Urban Areas against Investment Pressure. A Case Study of Romania and Poland” aims to identify potential ways to protect the environmental and natural values of urban areas from investment pressure in Romania and Poland, two counties with problematic management of such issues.

This was an engaging read, and the manuscript has merits and may be of great interest to the scientific community. However, it still needs polishing and a bit of restructuring/reorganisation. Also, it could be shortened, because it becomes a dull reading after a while. There are sections that need to be rewritten to be more on point, rather than present a lot of connected ideas that lead to a several basic research statements. My impression is that the paper has a lot of valuable content that just needs to be written in a straight forward manner, to make it more appealing to researchers. Also, the understanding of the research can be facilitated by adding several diagrams (detailed in comments).

Please note that I pointed the authors to some papers for documentation and for supporting problematic questions that may arise. These papers do no have to be cited in the paper, and I specify this precisely because I want to avoid any suspicions about requests to cite any research works.

Please see the comments in the attached pdf. I hope this review will contribute to the improvement of the manuscript, and that the authors will find the comments useful.

Author Response

Thank you for your comments, and for the time and efforts invested in the review process. Please find our response attached. Sincerely, the Authors.

Reviewer 2 Report

A very well written paper.

1. I would only recommend summarizing the work a little as it is very lengthy at the moment. Perhaps the section describing Poland and Romania could be summarized to give more focus on the content of the scientific work. 

2. A few layout edits can be done to enhance the overall look of the paper. For example, Tables could be adjusted to the text and images of the page as they currently extend to the left margin. The discussion could start on page 25 and so on. 

Well done. 

Author Response

(The authors gave the same response as above.)

Round 2

Reviewer 1 Report

I appreciate the effort put by the authors into improving the manuscript and addressing the comments in the last review. I would have preffered to see a new pdf with the implemented changes, not a clear version of the modified pdf. Perhaps this is a system error.

Here I will address only the parts that I consider that were not properly explained/argued by the authors (in case of disagreement) or that raised concerns from them. The suggestions that were not implemented by the authors, but where a sound explanation for their choice is provided are not to be addressed, as the explanation was considered satisfactory.

- However, Reviewer #1 requested additional details, and the two resulted into an introduction similar in size with the initial one. We are kindly asking Reviewer #1 to revisit this contradiction, and provide more specific comments concerning the paragraphs to be reduced.

I signaled to the authors that the Introduction was too long, and that it woulr benefit for some thorough editing (including shortening). Although long, the introduction was not complete. I see no contradiction here. The Introduction simply contained unuseful information and was lacking the important ones. The comparison the approaches of national spatial planning systems was too long and not vital. Reading such a vast introduction, the reader is prone to losing the stream of ideas and to missing the point of the paper. I strongly reccommend shortening the Background part and the Specific Issue part, aiming for a 900-word Introduction (or a 1.5 page one).

Please note that I provided the authors with instructions regarding what should improve the Introduction (contrary to their statement in Italics), but they argued against it.

- We have conceived the concluding section as a separate one, summarizing the rationale, methods, results, and discussions; we found a need for a “take home” message summarizing the main points. If we merge the two sections, the message will be obscured, and this was not our intention. We developed and reworded the conclusions, but insist on keeping them separate from the discussions.

My comment was misunderstood. I did nto advised for a union of Discussion and Conclusions. What I meant is that the current form of Conclusions better fits a Discussion section and it does not provide a critical view on the manuscript.

My final request for the authors is to shorten and improve the Introduction. Overall, I am pleased with the new version of the paper, but I still consider the Introduction is (not a little) too much.

Author Response

Thank you for your comments. For details, please refer to the attached cover letter.
